# Use of suboptimal control arms in randomized clinical trials of investigational cancer drugs in China, 2016–2021: An observational study

**Yichen Zhang**[1], **Dingyi Chen**[1], **Siyuan Cheng**[2], **Zhizhou Liang**[1], **Lu Yang**[2], **Qian Li**[2], **Lin Bai**[1], **Huangqianyu Li**[3], **Wei Liu**[4], **Luwen Shi**[1,3], **Xiaodong Guan**[1,3]*

1 Department of Pharmacy Administration and Clinical Pharmacy, School of Pharmaceutical Sciences, Peking University, Beijing, China, 2 Department of Medical Oncology and Radiation Sickness, Peking University Third Hospital, Beijing, China, 3 International Research Centre for Medicinal Administration, Peking University, Beijing, China, 4 Department of Pharmacy, Peking University Third Hospital, Beijing, China

* guanxiaodong@pku.edu.cn

**Data Availability Statement:** All relevant data are within the manuscript and its Supporting Information files.

**Funding:** XG received the National Natural Science Foundation of China award (Grant No. 72274004, http://www.nsfc.gov.cn/). The funders had no role

## Abstract

### Background

The use of suboptimal controls in randomized trials of new cancer drugs can produce potentially unreliable clinical efficacy results over the current standard of care and expose patients to substandard therapy. We aim to investigate the proportion of randomized trials of investigational cancer drugs that used a suboptimal control arm and the number of trial participants at risk of exposure to suboptimal treatments in China. The association between the use of a suboptimal control and concluding statistical significance on the primary endpoint was also examined.

### Methods and findings

This observational study included randomized controlled trials (RCTs) of cancer drugs that were authorized by specific Chinese institutional review boards between 2016 and 2021, supporting investigational new drug applications of these drugs in China. The proportion of trials that used a suboptimal control arm and the total number of trial participants at risk of exposure to suboptimal treatments were calculated. In a randomized trial for a specific condition, a comparator was deemed suboptimal if it was not recommended by clinical guidelines published in priori or if there existed a regimen with a higher level of recommendation for the indication.

The final sample included 453 Phase II/III and Phase III randomized oncology trials. Overall, 60 trials (13.2%) adopted a suboptimal control arm. Among them, 58.3% (35/60) used comparators that were not recommended by a prior guideline for the indication. The cumulative number of trial participants at risk of exposure to suboptimal treatments totaled 18,610 by the end of 2021, contributing 15.1% to the total number of enrollees of all sampled RCTs in this study. After adjusting for the year of ethical approval, region of participant

in study design, data collection and analysis, decision to publish, or preparation of the manuscript.

**Competing interests:** XG acknowledged receiving grants from the Research on Equitable Access and Rational Use of Medicines sponsored by the Peking University Health Science Center. XG and LS reported receiving grants from the National Natural Science Foundation of China outside the submitted work. The funders had no role in study design, data collection and analysis, decision to publish, or manuscript preparation. Other authors have declared that no competing interests exist.

**Abbreviations:** aOR, adjusted odds ratio; CI, confidence interval; CSCO, Chinese Society of Clinical Oncology; FDA, Food and Drug Administration; IRB, institutional review board; NCCN, National Comprehensive Cancer Network; NMPA, National Medical Products Administration; NSCLC, non-small cell lung cancer; RCT, randomized controlled trial.

recruitment, line of therapy, and cancer site, second-line therapies (adjusted odds ratio [aOR] = 2.7, 95%CI [1.2, 5.9]), adjuvant therapies (aOR = 8.9, 95% CI [3.4, 23.1]), maintenance therapies (aOR = 5.2, 95% CI [1.6, 17.0]), and trials recruiting participants in China only (aOR = 4.1, 95% CI [2.1, 8.0]) were more likely to adopt a suboptimal control. For the 105 trials with publicly available results, no statistically significant difference was observed between the use of a suboptimal control and concluding positive on the primary endpoint (100.0% [12/12] versus 83.9% [78/93], $p$ = 0.208). The main limitation of this study is its reliance on clinical guidelines that could vary across cancer types and time in assessing the quality of the control groups.

## Conclusions

In this study, over one-eighth of randomized trials of cancer drugs registered to apply for regulatory approval in China used a suboptimal comparator. Our results highlight the necessity to refine the design of randomized trials to generate optimal clinical evidence for new cancer therapies.

## Author summary

### Why was this study done?

- Using inappropriate controls in randomized trials of new cancer drugs can expose patients to harm and produce potentially unreliable results of clinical efficacy.

- In November 2021, China's Center for Drug Evaluation issued the *Guidance on Clinical Value-Oriented Oncology Drug Research and Development.* It stated that an active control arm should be selected with consideration of whether it reflects the optimal treatment in current clinical practices.

- Previous descriptive analysis showed that oncology trials are expanding in China; however, no literature evaluated the quality of their control arms.

### What did the researchers do and find?

- This observational study reports the annual trend and proportion of 453 Phase II/III and Phase III randomized oncology trials registered in China between 2016 and 2021 that adopted a suboptimal control arm. We also calculated the cumulative number of trial participants at risk of exposure to suboptimal treatments.

- We observed a fluctuating yet overall upward trend in the number of trials and participants that used a suboptimal comparator during our observation period. Overall, 60 trials adopted a suboptimal control arm and the cumulative number of trial participants at risk of exposure to suboptimal treatments totaled 18,610 by the end of 2021.

- Trials with a suboptimal control reported higher but statistically non-significant proportions of positive results than those using an optimal control.

**What do these findings mean?**

■ Trial sponsors, ethical review boards, and oncologists should make collaborative efforts to protect patients from unnecessary harm and drugs with uncertain clinical benefits over the existing standard of care.

■ Regulatory agencies should be cautious when reviewing investigational new drug applications whose supporting trial used a suboptimal control and when making subsequent market authorization decisions.

■ A limitation of this study is the reliance on clinical guidelines that could vary across cancer types and time in assessing the quality of the control groups. Hence, though oncologists and clinical pharmacists were consulted when assessing the sample trials, our results may have inherent subjectivity.

## Introduction

An investigational new drug has to demonstrate efficacy and establish its safety profile through clinical trials prior to receiving market approval, with randomized controlled trials (RCTs) considered as the "gold standard" for evaluating new drugs [1,2]. When designing an RCT, sponsors or investigators choose control groups prudently since they affect the inferences that can be drawn from the trial, the ethical acceptability of the trial, and the acceptability of the results by regulatory authorities [3]. Under the *Declaration of Helsinki*, the effectiveness of a new medical intervention, if applicable, must be evaluated against those of the best-proven interventions at present [4]. However, previous research has shown that 17% (16/97) of cancer drug approvals of the United States Food and Drug Administration (FDA) between 2013 and 2018 were supported by RCTs that used suboptimal controls [5]. That is, the control agent had not been recommended by a relevant guideline or had been demonstrated to be inferior to available therapy. Similarly, 7 of 49 (14%) RCTs investigating drugs for multiple myeloma in the US randomized participants to control arms despite the availability of alternative regimens proven to have superior clinical efficacy [6]. More recently, there has been growing concern about the selection of inappropriate control arms in trials of metastatic castrate-resistant prostate cancer [7, 8] and hematological malignancies [9–11].

Adopting a suboptimal control group in a cancer drug registration RCT may bias its results in favor of the experimental arm and compromise its probability of informativeness in evolving treatment landscape as the comparator does not reflect the current standard of care. Information from new cancer drug trials with an inferior control group would confound regulatory decision-making by regulators or oncologists [12,13]. Moreover, participants in RCTs are exposed to uncertain benefits and risks of investigational new therapies [14]. It was estimated that an FDA approval required more than 12,000 pre-license trial (also called the investigational new drug) participants and that around 110,000 patients were enrolled in trials that did not lead to an FDA approval [15]. Using suboptimal comparators in RCTs, especially unlicensed agents or regimens, can expose vulnerable patients to unnecessary substandard therapy and produce unconvincing comparative efficacy results of new cancer drugs over the existing standard of care [16]. Of the 387 oncology RCTs published in 11 major journals between 2017 and 2021, 43 (11%) were deemed to have a suboptimal control, with a total of 9,505 patients assigned to the suboptimal treatment arms [17]. At the core of considerations in an ethical review are principles of respect for persons, beneficence (do not harm and maximize possible

benefits and minimize possible harms), and justice [18], hence, research resources could be more effectively utilized if institutional review boards (IRBs) embrace their responsibilities by ensuring that each trial receives meaningful scientific review [19].

The past decade witnessed a large number of oncology trials [20,21] and new cancer drug approvals in China [22,23]. Since 2016, more than 200 cancer drug clinical trials have been initiated in mainland China annually [20]. In November 2021, the Centre for Drug Evaluation at China's National Medical Products Administration (NMPA) issued the *Guidance on Clinical Value-Oriented Oncology Drug Research and Development*, it suggests that an active control arm should be selected with consideration of whether it reflects the optimal treatment in current clinical practices [23,24]. There is no empirical evidence, however, about the appropriateness of the control arm of these cancer drug clinical trials. Moreover, the association between the quality of the control arm and trial results was rarely scrutinized. In this observational study, we aim to identify randomized oncology registration trials using suboptimal control arm in China and estimate the number of trial participants at risk of receiving suboptimal treatments. For RCTs with published results, we further investigated the proportion of trials with statistically significant results on their primary endpoint and examined the association between the adoption of a suboptimal control arm and concluding statistical significance.

## Methods

### Data sources

We obtained clinical trials registered on China's NMPA Registration and Information Disclosure Platform for Drug Clinical Studies by December 31, 2020 [25]. This publicly available website was launched in 2012. In September 2013, China's Food and Drug Administration (now known as the NMPA) required that all clinical trials (including bioequivalence trials, pharmacokinetics trials, Phase I, II, III, and IV trials) authorized by the agency for investigational new drug applications should be registered on this platform [26]. Trial information, including the trial identifier, drug name, indication, sponsor, location, trial phase, study design, treatment, eligibility criteria, endpoints, institutions, ethical approval date, start date, status, and enrollment, are disclosed and made publicly accessible on the website [20].

We referred to Chinese oncology guidelines to determine whether the trials in our study sample used a suboptimal control arm. We first examined guidelines published by the Chinese Society of Clinical Oncology (CSCO). CSCO started publishing guidelines for various cancer types in 2016 and has been updating them annually or every other year. The development of CSCO Guidelines follows the level of evidence and expert consensus [27]. CSCO categorizes evidence-based recommendations into Grade I (Evidence level 1A and some Evidence level 2A), II (Evidence level 1B and some Evidence level 2A), and III (Evidence level 2B and 3). For cancer types without a specific CSCO guideline, we searched clinical practice guidelines published by other academic societies listed in the China National Knowledge Infrastructure and/ or the WanFang Database (see **Box A in S1 Text** for search terms and strategies). The review adhered to the Preferred Reporting Items for Systematic Reviews and Meta-Analyses (PRISMA) guideline (see **S1 PRISMA Checklist**). This study did not involve patient data, and, thus, the ethical approval was waived by the IRB of Peking University Health Science Center. Written informed consent was not required.

### Sample selection

From all clinical trials registered on the NMPA platform, we used a prespecified list of search terms in Chinese (see **Box B in S1 Text** for further details) to identify Phase II/III and III oncology RCTs conducted among adult patients and approved by any IRB between January 1,

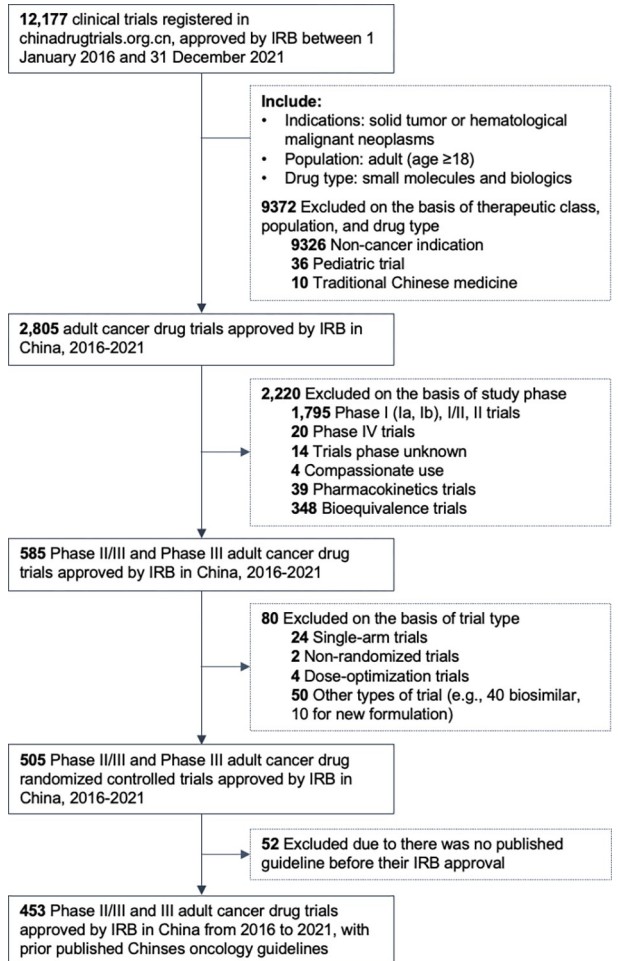

**Fig 1. Flowchart of sample oncology randomized clinical trials selection. Abbreviation:** IRB, institutional review board.

2016 and December 31, 2021. We included trials investigating small-molecule entities (i.e., chemotherapeutic agents and molecule-targeted drugs) and biologics. Trials investigating traditional Chinese medicines, diagnostic products, and prophylactic vaccines were excluded (**Fig 1**). For each trial, we extracted data on its specific indication(s), inclusion and exclusion criteria, intervention, and control arm for analysis. Trials were categorized by the year of first ethical approval, study phase, cancer site, line of therapy, applicant type (China's domestic versus multinational company), and region of participant recruitment (international multicenter versus China-only).

## Definition of suboptimal control arm

For each trial in our sample, we defined the quality of the control arm as suboptimal if **(1)** the control was not recommended by the concerned clinical practice guideline for a specific indication issued prior to the trial's IRB approval (i.e., different from the regimen recommended in guidelines published before the trial's IRB approval); **(2)** the control drug or regimen was specified in the guideline but not the recommended regimen (e.g., the control arm was a single agent when the guideline published before the trial's IRB approval recommended doublet

therapy); and **(3)** the control arm was among the recommended list of therapies but there existed a drug (or drugs) with a higher level of recommendation (e.g., the control regimen had a Grade II recommendation while an alternative drug had a Grade I recommendation; see **Box 1** for more details).

---

### Box 1. Criteria used to judge the quality of a control arm in randomized controlled trials to be suboptimal

➢ **Category 1. The control was not recommended by the concerned clinical practice guideline for a specific indication issued prior to the trial's ethical approval**

e.g., In CTR20160389, a second-line therapy for HER2-positive breast cancer progressed after chemotherapy, the comparator was placebo in combination with capecitabine. According to the *Guideline for Diagnosis and Treatment of Breast Cancer (2015 Edition)*, for HER-2-positive advanced, recurrent and metastatic breast cancer, the preferred treatment should be anti-HER-2 therapy.

➢ **Category 2. The control drug or regimen was specified in the guideline but not the recommended regimen**

e.g., In CTR20180332, a first-line therapy for biliary tract carcinoma, the comparator was gemcitabine in combination with oxaliplatin. According to the *Guideline for Diagnosis and Treatment of Gallbladder Carcinoma (2015 Edition)*, gemcitabine in combination with cisplatin was recommended as the standard of care. In the later published *Chinese Society of Clinical Oncology for Biliary Tract Cancer (2021 Edition)*, gemcitabine in combination with oxaliplatin only got a Grade of Recommendation II.

➢ **Category 3. The control arm was among the recommended list of therapies, but there existed a drug (or drugs) with a higher level of recommendation**

e.g., In CTR20171558, a second-line therapy for HER2-negative, estrogen receptor-positive breast cancer progressed after nonsteroidal aromatase inhibitors, the comparator was placebo in combination with exemestane. According to *Chinese Society of Clinical Oncology Guideline for Breast Cancer (2017 Edition)*, fulvestrant was recommended as the basic treatment strategy, while aromatase inhibitors in another mechanism of action were recommended as an alternative.

**Abbreviation:** HER2, human epidermal growth factor receptor 2.

---

For multinational trials, in consideration of the delay in new drug approval between China and other countries [28], we further reviewed the Drugs@FDA database [29] to check whether the control regimen had been approved in the US, where most new cancer therapies were approved first [30], before its date of IRB approval in China. If a multinational trial encompassed a specific indication on which a treatment consensus had not been reached in China, we checked whether there existed a standard of care in the former National Comprehensive Cancer Network (NCCN) Guidelines, which have a worldwide influence on clinical practice [31], at that time. Moreover, given the time lag in updating clinical guidelines [32], we also investigated whether a potentially suboptimal control regimen was deemed optimal in any revised guidelines later. Regimens that had been approved by the FDA, recommended as a standard-of-care in the NCCN guideline, or recommended in subsequent Chinese guidelines were recategorized as "potentially optimal control."

Two investigators (D.C. and Z.L.) independently extracted information about the indication and eligibility criteria of all RCTs and matched corresponding guideline recommendations for further investigation. The initial agreement rate achieved 94.9% (*n* = 430/453). Another investigator (Y.Z.) resolved disagreements. Three oncologists (S.C., L.Y., and Q.L.) and an oncology pharmacist (W.L.) independently reviewed the control arm to identify suboptimal control arms. Disagreements were resolved by consensus.

## Identification of publication of sample trials

For each trial, we used a stepwise approach to systematically search for peer-reviewed publications of its results [33], given the results submitted to the NMPA Registration and Information Disclosure Platform for Drug Clinical Studies are not publicly available. First, we searched ClinicalTrials.gov (the US National Library of Medicine database of clinical trials) using Study IDs (or Study Title/Acronym when necessary) to match with their National Clinical Trial number. Second, we searched MEDLINE (via PubMed) using the National Clinical Trial number and reviewed all corresponding publications. If no publication was found through these 2 steps, we further searched PubMed for RCTs using a precision-maximizing search strategy and a combination of terms including the drug's name (if unavailable, drug code) and indication [22].

We reviewed the titles and abstracts of all records to identify the primary publications reporting trial results. Among all eligible publications, we extracted information about a trial's primary endpoints (i.e., statistical analysis plan, trial results, and statistical significance). If more than one publication reported trial results, we relied on the initial one that reported the preplanned analysis of the primary endpoint(s). We also examined the availability of a pre-specified statistically significant benefit of the sample RCTs. For trials set coprimary endpoints, the results were deemed as statistically significant if the difference between the experimental and the control group was considered to be statistically significant for at least one endpoint.

## Outcome measures

The primary outcomes were the annual proportion of trials with a suboptimal control arm and the number of trial participants exposed to suboptimal treatments. We calculated the annual and cumulative number of planned enrollees of sample RCTs with a suboptimal control arm in mainland China. For trials that did not disclose their planned sample size, we relied on the actual number of participants enrolled as of 31 August 2022. The secondary outcomes were the potential predictors of the use of suboptimal control, and, for trials that have available results, the association between the adoption of suboptimal control and concluding statistical significance for the primary endpoint(s).

## Data analysis

We used descriptive statistics to characterize the proportion of RCTs that used a suboptimal control arm for registration purposes in China, stratified by the year of their first ethical approval. To identify potential predictors of the use of suboptimal control, the univariable logistic regression, chi-squared test, and Fisher's exact test were performed to test differences between groups in categorical variables. Two-sided *P* values of $< 0.05$ were considered statistically significant. Factors found to be significant in univariable analysis were adjusted in the multivariable logistic regression model to examine subgroups with high odds of using a suboptimal control. All models were computed using STATA MP version 14.0 (StataCorp). Data were analyzed from October 2022 to September 2023.

## Results

### Characteristics of sample oncology trials and clinical practice guidelines

Fig 1 illustrates the process of trial identification. From January 1, 2016 to December 31, 2021, a total of 2,805 cancer drug trials for adult patients were approved by any IRB in mainland China. Based on the inclusion criteria, we included 453 Phase II/III and Phase III cancer drug RCTs. Of these, 78.6% (356/453) were compared against 54 CSCO guidelines to determine whether RCTs in our sample adopted a suboptimal control arm. For the other 21.4% (97/453) trials without a specific CSCO guideline available prior to their first ethical approval, we referred to a total of 29 clinical guidelines published by other Chinese academic societies (see **Table A in S1 Text** for further details).

Table 1 shows the characteristics of the 453 eligible RCTs, which included 21 Phase II/III trials (4.6%) and 432 Phase III trials (95.4%). The annual number of cancer drug RCTs for registration purposes increased from 20 (4.4% of 453 eligible registration trials in the study time period) in 2016 to 130 (28.7%) in 2021. Around half of the trials (228/453, 50.3%) were sponsored by China's domestic companies. More than half of the trials (241/453, 53.2%) were indicated for first-line therapy, followed by second-line (105/453, 23.2%) and adjuvant therapy (39/453, 8.6%). Trials were mostly for the treatment of cancers of the lung (143/453, 31.6%), breast (68/453, 15.0%), stomach (34/453, 7.5%), liver (33/453, 7.3%), and prostate (25/453, 5.5%). Around half of the trials (251/453, 55.4%) were international multicenter studies, while 202 (44.6%) recruited patients from mainland China only. Across all 453 trials, the planned median number of participants in mainland China was 220 (interquartile range: 100 to 396 participants).

### Oncology RCTs using a suboptimal control arm

Of the 453 eligible RCTs, the control arm of 369 (81.5%) trials was categorized as optimal. The control arm of 24 (5.3%) trials was categorized as "potentially optimal," including 41.7% (10/24) multinational trials whose control group had been approved by the US FDA before receiving IRB approval in China, 25.0% (6/24) trials whose control group were cited as the global standard-of-care by the NCCN, and 33.3% (8/24) trials whose control regimens were later recommended by Chinese guidelines (see **Table B** in S1 Text for the details).

A total of 13.2% of RCTs (60/453) adopted a suboptimal control. The control treatment was not recommended by the relevant guideline in 36 cases (60.0%; Table 2); there was an alternative drug with a higher level of guideline recommendation in 21 cases (35.0%); and the control drug was not given in the recommended treatment combination in 4 cases (6.7%; see **Table C in S1 Text** for details). From 2016 to 2021, the annual proportion of trials that used a suboptimal control arm of all oncology registration trials fluctuated between 7.0% (3/40) and 20.0% (26/130; Fig 2A).

The annual and cumulative number of trial participants at risk of exposure to suboptimal treatments are presented in Fig 2B. From 2016 to 2021, a total of 123,452 cancer patients were planned to be enrolled in the sample RCTs. The annual number of participants who would be assigned to receive suboptimal treatments in China increased over time. From 510 in 2016 and reached 7,476 in 2021. By the end of 2021, the cumulative number of trial participants in mainland China at risk of exposure to suboptimal treatments totaled 18,610, taking 15.1% (18,610/123,452) of the total number of all sample RCT enrolments.

### Availability and statistical significance of trial results

Of 1,132 publications obtained from MEDLINE by September 2023 (median follow-up duration since the IRB approval: 3.4 years, interquartile range: 2.4 to 4.3 years), we identified 131 reporting

**Table 1. Characteristics of and potential factors associated with adoption of suboptimal control arm of randomized trials for investigational cancer drug applications in China.**

| Characteristics | All trials, Number (%) (N = 453) | Trials, Number (%) | | Unadjusted OR (95% CI)[a] | P value of univariable analysis[b] | Adjusted OR (95% CI) of multivariable analysis[c] |
| --- | --- | --- | --- | --- | --- | --- |
| | | Optimal and potential optional control arm (N = 393 [369 + 24], 86.8%) | Suboptimal control arm (N = 60, 13.2%) | | | |
| **Year of ethical approval** | | | | | | |
| 2016 | 20 (4.4) | 17 (85.0) | 3 (15.0) | 2.4 (0.4, 12.9) | 0.160 | 1.7 (0.3, 11.1) |
| 2017 | 43 (9.5) | 40 (93.0) | 3 (7.0) | Reference | | Reference |
| 2018 | 73 (16.1) | 65 (89.0) | 8 (11.0) | 1.6 (0.4, 6.5) | | 1.4 (0.3, 6.2) |
| 2019 | 99 (21.9) | 87 (87.9) | 12 (12.1) | 1.8 (0.5, 6.9) | | 1.3 (0.3, 5.6) |
| 2020 | 88 (19.4) | 80 (90.9) | 8 (9.1) | 1.3 (0.3, 5.3) | | 1.1 (0.2, 4.7) |
| 2021 | 130 (28.7) | 104 (80.0) | 26 (20.0) | 3.3 (1.0, 11.6) | | 3.0 (0.8, 11.7) |
| **Applicant** | | | | | | |
| China's domestic company | 228 (50.3) | 193 (84.6) | 35 (15.4) | Reference | 0.183 | |
| Multinational company | 225 (49.7) | 200 (88.9) | 25 (11.1) | 0.7 (0.4, 1.2) | | |
| **Phase** | | | | | | |
| II/III | 21 (4.6) | 19 (90.5) | 2 (9.5) | Reference | 0.457 | |
| III | 432 (95.4) | 374 (86.6) | 58 (13.4) | 1.5 (0.3, 6.5) | | |
| **Recruitment region** | | | | | | |
| International multicenter | 251 (55.4) | 232 (92.4) | 19 (7.6) | Reference | <0.001 | Reference |
| China-only | 202 (44.6) | 161 (79.7) | 41 (20.3) | 3.1 (1.7, 5.6) | | 4.1 (2.1, 8.0) |
| **Line of therapy** | | | | | | |
| First-line | 241 (53.2) | 225 (93.4) | 16 (6.6) | Reference | <0.001 | Reference |
| Second-line | 105 (23.2) | 85 (81.0) | 20 (19.0) | 3.3 (1.6, 6.7) | | 2.7 (1.2, 5.9) |
| ≥Third-line | 21 (4.6) | 19 (90.5) | 2 (9.5) | 1.5 (0.3, 6.9) | | 1.2 (0.2, 6.2) |
| Adjuvant | 39 (8.6) | 27 (69.2) | 12 (30.8) | 6.3 (2.7, 14.6) | | 8.9 (3.4, 23.1) |
| Maintenance | 24 (5.3) | 17 (70.8) | 7 (29.2) | 5.8 (2.1, 16.0) | | 5.2 (1.6, 17.0) |
| Neoadjuvant + adjuvant | 12 (2.6) | 10 (83.3) | 2 (16.7) | 2.8 (0.6, 13.9) | | 4.5 (0.8, 25.7) |
| neoadjuvant | 11 (2.4) | 10 (90.9) | 1 (9.1) | 1.4 (0.2, 11.7) | | 1.1 (0.1, 10.0) |
| **Type of therapy** | | | | | | |
| Combination therapy | 324 (71.5) | 285 (88.0) | 39 (12.0) | Reference | 0.403 | |
| Monotherapy | 118 (26.0) | 99 (83.9) | 19 (16.1) | 1.6 (0.3, 7.8) | | |
| Mono/combination therapy (3-arm) | 11 (2.4) | 9 (81.8) | 2 (18.2) | 1.4 (0.8, 2.5) | | |
| **Cancer site** | | | | | | |
| Lung | 143 (31.6) | 129 (90.2) | 14 (9.8) | 2.6 (0.3, 20.7) | 0.018 | 1.5 (0.2, 12.9) |
| Breast | 68 (15.0) | 53 (77.9) | 15 (22.1) | 6.8 (0.8, 54.4) | | 3.3 (0.4, 28.7) |
| Stomach | 34 (7.5) | 33 (97.1) | 1 (2.9) | 0.7 (0.0, 12.2) | | 0.4 (0.0, 7.0) |
| Liver | 33 (7.3) | 29 (87.9) | 4 (12.1) | 3.3 (0.3, 31.6) | | 1.7 (0.2, 17.7) |
| Prostate | 25 (5.5) | 24 (96.0) | 1 (4.0) | Reference | | Reference |
| Lymphoma | 24 (5.3) | 21 (87.5) | 3 (12.5) | 3.4 (0.3, 35.5) | | 2.6 (0.2, 29.2) |
| Esophagus | 18 (4.0) | 15 (83.3) | 3 (16.7) | 4.8 (0.5, 50.5) | | 3.4 (0.3, 39.2) |
| Leukemia | 15 (3.3) | 14 (93.3) | 1 (6.7) | 1.7 (0.1, 29.6) | | 2.7 (0.1, 50.7) |
| Multiple myeloma | 13 (2.9) | 10 (76.9) | 3 (23.1) | 7.2 (0.7, 77.8) | | 5.6 (0.5, 68.5) |
| Ovary | 12 (2.6) | 7 (58.3) | 5 (41.7) | 17.1 (1.7, 172.1) | | 5.5 (0.5, 66.6) |
| Other | 68 (15.0) | 58 (85.3) | 10 (14.7) | 4.1 (0.5, 34.1) | | 2.3 (0.3, 20.4) |

*(Continued)*

**Table 1.** (Continued)

| Characteristics | All trials, Number (%) (N = 453) | Trials, Number (%) | | Unadjusted OR (95% CI)[a] | P value of univariable analysis[b] | Adjusted OR (95% CI) of multivariable analysis[c] |
|---|---|---|---|---|---|---|
| | | Optimal and potential optional control arm (N = 393 [369 + 24], 86.8%) | Suboptimal control arm (N = 60, 13.2%) | | | |
| **Preplanned participants, Number** | | | | | | |
| <100 | 108 (23.8) | 100 (92.6) | 8 (7.4) | Reference | 0.345 | |
| 100–199 | 102 (22.5) | 89 (87.3) | 13 (12.7) | 1.8 (0.7, 4.6) | | |
| 200–299 | 61 (13.5) | 50 (82.0) | 11 (18.0) | 2.8 (1.0, 7.3) | | |
| 300–399 | 70 (15.5) | 60 (85.7) | 10 (14.3) | 2.1 (0.8, 5.6) | | |
| 400–499 | 50 (11.0) | 43 (86.0) | 7 (14.0) | 2.0 (0.7, 6.0) | | |
| ≥500 | 62 (13.7) | 51 (82.3) | 11 (17.7) | 2.7 (1.0, 7.1) | | |
| **Data and Safety Monitoring Board** | | | | | | |
| Has | 301 (66.4) | 267 (88.7) | 34 (11.3) | Reference | 0.085 | |
| None | 152 (33.6) | 126 (82.9) | 26 (17.1) | 1.6 (0.9, 2.8) | | |

CI, confidence interval; OR, odds ratio.

[a]Unadjusted ORs were calculated using logistic regression.

[b]P values were obtained from the univariable logistic regression, chi-squared test, and Fisher's exact test.

[c]Adjusted ORs were obtained from the multivariable logistic regression model. Factors identified in univariable analysis were adjusted: year of ethical approval, recruitment region, line of therapy, and cancer site.

the preplanned results of 105 sample trials (23.2%). Among them, 105 reporting the initial trial results were included for data extraction. Trials mostly used progression-free survival (56/105, 53.3%) and overall survival (24/105, 22.9%) as the primary endpoint. Some studies (14/105, 13.3%) used overall survival and progression-free survival concomitantly as the primary endpoints (see **Table D in S1 Text**). Of the 93 trials that were deemed to have used an optimal or potentially optimal control, 78 (83.9%) reported statistically significant results while 15 (16.1%) concluded statistical nonsignificance (**Table 3**; see **Table E in S1 Text** for details). Of the trials that were deemed as having a suboptimal control, all (12/12, 100.0%) reported statistically significant results. No statistically significant association was observed between the use of suboptimal control and concluding statistical significance on the primary endpoint (p = 0.208).

## Subgroup analysis

The univariate analysis showed that the use of a suboptimal control varied across RCTs investigating drugs of different lines of therapies (p < 0.001; Table 1). Trials' recruitment location

**Table 2. Suboptimal control type, by applicable guidelines issued by various societies.**

| Suboptimal Control Type | Trials, Number (%) (N = 60) | Trials, Number (%) | |
|---|---|---|---|
| | | Applicable to guidelines issued by Chinese Society of Clinical Oncology (N = 49) | Applicable to guidelines issued by other academic societies (N = 11) |
| The comparator was not recommended by the guideline for specific indication | 35 (58.3) | 27 (55.1) | 8 (72.7) |
| The comparator was not the specifically recommended regimen | 4 (6.7) | 2 (4.1) | 2 (18.2) |
| The comparator was recommended, but there existed higher-level recommendation | 21 (35.0) | 20 (40.8) | 1 (9.1) |

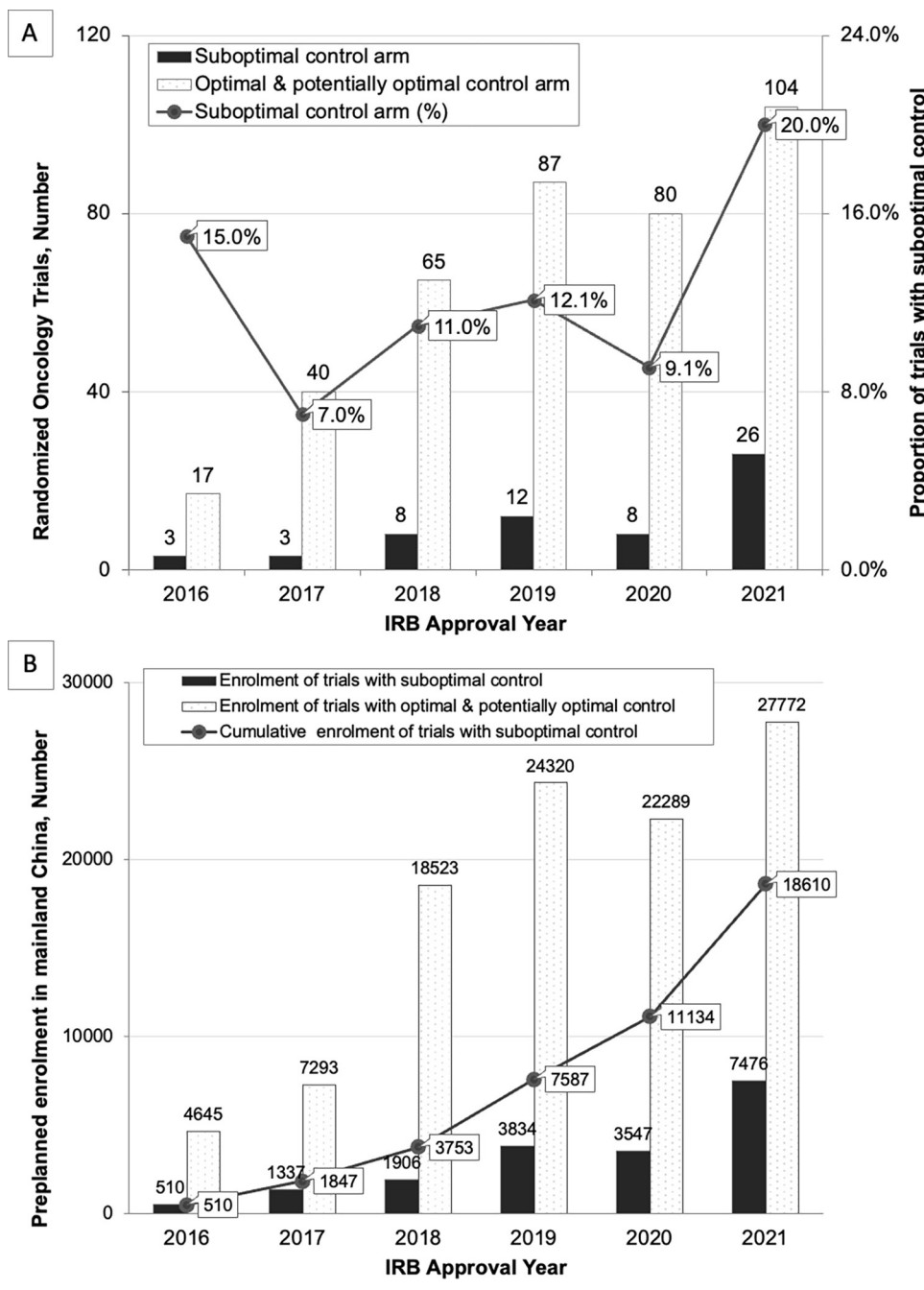

**Fig 2. Temporal trends of registrational randomized clinical trials of new cancer drugs used suboptimal control arms.** (**A**) Annual numbers of oncology randomized clinical trials that used a suboptimal control arm in mainland China. (**B**) Annual and cumulative numbers of patients exposed to suboptimal treatment of oncology randomized clinical trials in mainland China. Abbreviation: IRB, institutional review board. Note: For the 5 trials that did not disclose their plan of patient enrolment, we used the actual number of participants as of 31 August 2022 alternatively.

(international multicenter versus China-only, $p < 0.001$) and cancer site ($p = 0.018$) were also identified as potential predictors of using a suboptimal control. After adjusting these factors, multivariable logistic regression showed that second-line therapies (adjusted odds ratio (aOR) = 2.7, 95% CI [1.2, 5.9]), adjuvant therapies (aOR = 8.9, 95% CI [3.4, 23.1]), maintenance

**Table 3. Availability and statistical significance of sample randomized oncology trial results.**

| Primary Endpoint Results | All, Number (%) | Trials, Number (%) | | P value[a] |
|---|---|---|---|---|
| | | Optimal and potentially optimal control arm | Suboptimal control arm | |
| **Total** | **453 (100.0)** | **393 (100.0)** | **60 (100.0)** | |
| *Available* | *105 (23.2)* | *93 (23.7)* | *12 (20.0)* | |
| Concluding statistical significance[b] | 90 (85.7) | 78 (83.9) | 12 (100.0) | 0.208 |
| Concluding statistical nonsignificance[b] | 15 (14.3) | 15 (16.1) | 0 | |
| *Unavailable* | *348 (76.8)* | *300 (76.3)* | *48 (80.0)* | |

[a]*P* value was obtained from Fisher's exact test.

[b]Of trials with publicly available results.

therapies (aOR = 5.2, 95% CI [1.6, 17.0]), and trials recruited in China only (aOR = 4.1, 95% CI [2.1, 8.0]) were more likely to adopt a suboptimal control. Although it was challenging to conduct a direct comparison when referring to different guidelines, the proportion of trials adopting a suboptimal control was similar when being compared against CSCO guidelines (49/356, 13.8%) or guidelines published by other societies (11/97, 11.3%, *p* = 0.614; **Fig A in S1 Text**).

## Discussion

In our analysis of 453 Phase II/III-III RCTs of new cancer drugs for registration purposes in mainland China in 2016 to 2021, we found that 13.2% of them adopted a suboptimal control arm. Approximately half of these were regimens not recommended by previously published clinical guidelines. The annual number of Chinese cancer participants who were randomized to receive suboptimal treatments in clinical trials totaled 18,610 by the end of 2021. Trials with suboptimal control reported higher but statistically nonsignificant proportions of positive results than those used optimal control.

The use of suboptimal or inferior control in oncology trials has been increasingly concerning in recent years [5–7,10,11,34]. Per the *Declaration of Helsinki*, clinical trials, whether initiated by an investigator or for registration purposes, should be conducted under prior ethical approval [4]. Under Part 312 of the Code of Federal Regulations and FDA Amendments Act of 2007, the US FDA issued regulatory requirements for the control of investigational drug [35]. Yet, evidence showed that 17% to 25% of superiority RCTs supporting FDA approval of cancer drugs in 2013 to 2019 used a substandard control arm [5,34]. Similarly, we found that approximately one-eighth of RCTs for new cancer drug applications in China between 2016 and 2021 were conducted using a suboptimal comparator. Adoption of a control regimen known to be inferior to available therapies or not recommended by specific guidelines was the primary contributor, as found in this research and in the literature [5]. Although it is challenging to compare regulatory outcomes across different settings and time periods, it seems that the proportion of RCTs registered in China with an optimal control was comparable to those in other developed countries, which suggests that trials registered in China are no less aligned with contemporary ethical standards than are in the US or European countries.

Adoption of a suboptimal control in RCTs is also prevalent in various diseases apart from oncology, such as rheumatology and cardiology [36,37]. It could be explained from the perspectives of study investigators, regulators, and the landscape change of disease treatment. Since large-scale multicenter oncology trials are costly and entail a high risk of failure, trial sponsors are not always incentivized to conduct head-to-head studies of their product with the best-proven therapy [5]. In our sample of RCTs, several investigated combination therapies

against a control that has the same mechanism of action as the standard-of-care drug (e.g., almonertinib [an EGFR tyrosine kinase inhibitor] for untreated advanced EGFR mutation-positive non-small cell lung cancer [NSCLC]). A possible justification for this may be that authorized agents within a therapeutic class may have comparable effectiveness. Nonetheless, not comparing an investigational regimen against a proven best therapy generates suboptimal evidence. Our results also showed that compared with trials using optimal control, those adopting a suboptimal control showed a higher proportion of concluding positive results, which favored the pharmaceutical companies, though this was not statistically significant, and only 23.2% of sample trial results were publicly available with limited follow-up time. It was noteworthy that 83.9% (78/93) and 100.0% (12/12) of our sample RCTs with optimal and suboptimal control concluded positive primary endpoint(s) respectively, which was higher than those (51.4% [177/344] and 76.7% [33/43] of solid tumor trials published among 11 major oncology journals) reported in a recent literature [11]. Given that this study focused on randomized oncology registration trials in China, one of the concerns is the selective reporting of results, geared toward those positive, may cause patients and regulators to exaggerate the clinical benefits of the investigational drugs. If this practice is widespread in the development pipeline, it may hamper therapeutic innovation and threaten the promotion of optimal clinical practice in the long term [8].

We observed a fluctuated but overall upward trend in the annual proportion of sample trials that used a suboptimal control. Compared with international multicenter RCTs, trials that recruited participants only in China seemed more likely to adopt an inferior control. A potential explanation is that trials initiated by multinational companies are more likely to recruit participants in high-income countries, and, thus, the proportion of their trials using a suboptimal control might be underestimated [38]. Globally, with the emergence of new drugs replacing existing standard-of-care therapies and the update of clinical practice guidelines, the optimal therapy also changes over time and may differ across regions [39,40]. Investigational new drug applications whose trials used a suboptimal comparator may distort regulators' decision-making and result in a potential waste of resources. In February 2022, the US FDA Oncologic Drugs Advisory Committee rejected the biologic license application of sintilimab for the first-line treatment of NCSLC. One of the committee's considerations was that its RCT used chemotherapy, rather than US-authorized immunotherapy, as the control arm [41]. Building consensus on the selection of an appropriate comparator in oncology registrational RCTs is therefore important. Although China's *Guidance on Clinical Value-Oriented Oncology Drug Research and Development* issued in November 2021 emphasized patient-centered cancer drug research and development and pointed out that an active control arm should be selected with considerations of whether it reflects the optimal treatment in current clinical practices, its definition of "optimal treatment" is ambiguous [24]. Therefore, despite that China's *Medicinal Product Administration Law* requires IRB approval and the regulatory agency's licensing for all registrational trials [42], the lack of high-level and harmonized criteria against which ethical approval can be made may allow for flawed designs in clinical trials.

The subgroup analysis, after adjusting for covariates, showed that adjuvant (mainly for lung cancer and hepatocellular carcinoma) and maintenance trials (mainly for ovarian cancer) were more likely to use suboptimal control. These findings could be explained by the evolving treatment landscape and discrepancies between that in China and in other regions. For high-risk Stage IB NSCLS, China's latest clinical practical guideline did not recommend adjuvant chemotherapy due to the absence of high-level evidence, while the NCCN Guideline recommended observation or adjuvant chemotherapy/osimertinib (for EGFR exon 19 deletion or L858R). For hepatocellular carcinoma with high risk of recurrence, adjuvant transarterial chemoembolization after curative resection got a Grade-I recommendation in China's guideline

(Level-of-evidence: 2A) with documented survival gains [43], while there was no global consensus on the treatment being a standard of care. For BRCA1/BRCA2 mutation-positive platinum-sensitive epithelial ovarian cancer, though poly ADP-ribose polymerase inhibitor has been recommended as the maintenance therapy since 2019, still, several placebo-controlled trials on next-in-class poly ADP-ribose polymerase inhibitor have been initiated subsequently.

Our findings also have important clinical implications. When a standard of care or an alternative treatment known to have superior benefit is available, being randomized into a suboptimal control arm may cause a patient to miss optimal care. Enrolling patients, especially key subpopulations, into a suboptimal control arm may also imply a waste of potential participant pool that could otherwise generate evidence to support future care. To address this issue, trial sponsors and investigators should shoulder the primary responsibility to ensure patient safety and trial validity. Clinicians should fully inform the trial participants about the benefits and risks entailed in their participation, not only of the investigational new drug but also of the authorized regimen in the control group. Moreover, although we observed a statistically non-significant effect of containing a Data Safety Monitoring Board on the adoption of suboptimal control in the multivariable model, setting a Data Safety Monitoring Board is also a feasible strategy to halt a clinical trial early and protect participants from receiving investigational drugs that were proven to lack therapeutic value in the interim analysis [44].

The strengths of our study are that we examined the quality of control arms of randomized trials evaluating the efficacy of investigational new cancer drugs, regardless of whether they have received market authorization, in this new era of China's oncology innovation. However, this study yields several limitations. **First**, 52 RCTs were excluded as there were no previously published Chinese guidelines to compare their controls against. Thus, the proportion of trials categorized as having an "optimal" control in this study might be conservative. **Second**, we used guidelines published prior to the date of a trial's initial ethical approval to evaluate the quality of their control arms. Some guidelines may not be available to some IRBs in a timely manner. However, among RCTs that used a suboptimal control, most (46 trials, 76.7%; **Table F in S1 Text**) were subject to clinical guidelines published 3 months or more before their ethical approval. **Third**, some guidelines may be subject to flaws, such that they were not updated timely or they expanded to include some options that were actually not relevant to the population being studied [45]. In some CSCO Guidelines, treatments that are expensive but may have substantial benefits for the patients are also regarded as Grade II recommendations, which may bias the results on Category 3. Besides, although we used FDA approvals and NCCN recommendations to reevaluate the quality of control arm of multinational trials, clinical benefits and level of evidence supporting their approval and recommendation may have discrepancies and deficiencies [46,47]. To mitigate these impacts, we invited several oncologists and clinical pharmacists to cross-check if our results were in line with their observations in actual clinical practices. **Fourth,** guidelines issued by expert committees on different specializations were developed according to varied principles, which might vary across types and time. This might bias the results of the univariable analysis. **Fifth**, we relied on the numbers of planned enrolment to estimate the number of trial participants at risk of receiving suboptimal treatments as two-thirds of the sample trials did not report the actual enrollment by the end of our observation. These numbers may deviate from the numbers of final trial recruitment. **Sixth**, publicly available data may represent a selective subset of trial information since only 23.2% of sample trial results were published by the end of our observation. **Seventh**, we only evaluated the quality of the control arm of RCTs and thus did not analyze the potentially inappropriate use of single-arm designs for conditions with existing standards of care. With the increasing number of nonrandomized oncology trials in China [22], further attention should be given to the rationality of using single-arm trials as the pivotal evidence for regulatory

approval. Similarly, we only assessed the regimen rather than the dose or schedule of comparator arms. Further study may be necessary to evaluate the quality of the latter if a standard dose or schedule is built. **Eighth**, we only included trials conducted among adults due to the limited number of Phase III childhood cancer trials (only 4) and the poor availability of guidelines concerned with childhood cancer in China before 2021. **Ninth**, we only evaluated cancer drug RCTs, and, thus, the results may not be applicable to other types of therapeutic agents. **Lastly**, the study protocol and analyses were not prespecified in an open database.

## Conclusions

Approximately one-eighth of randomized trials of cancer drugs for registration purposes in mainland China adopted a suboptimal control arm, and the number of trial participants at risk of receiving suboptimal cancer treatments in control arms increased over time. With the rising number of oncology trials registered in China, regulators should be cautious when making market authorization decisions for investigational new drugs that adopt suboptimal control in their pivotal trials.

## Supporting information

**S1 PRISMA Checklist. PRISMA 2020 checklist.**
(DOCX)

**S1 Text. Supporting information.** Box A. Key words and research strategy to identify clinical practice guidelines other than Chinese Society of Clinical Oncology. Box B. Key words and research strategy to identify oncology trials. Table A. Characteristics of oncology guidelines used to evaluate the quality of trial comparator. Table B. Randomized oncology clinical trials with a potentially optimal control arm in China, 2016–2021. Table C. Randomized oncology clinical trials with a suboptimal control arm in China, 2016–2021. Table D. Primary endpoints used in sample randomized oncology trials with published results. Table E. Primary endpoint results and statistical significance of sample randomized oncology trials. Fig A. Use of suboptimal control arm in randomized controlled oncology trials in China, by applicable guideline type. Table F. Guideline published duration before the ethical approval of sample randomized controlled trials.
(DOCX)

## Acknowledgments

We thank Professor Huseyin Naci from the Department of Health Policy, London School of Economics and Political Science, Professor Anita K. Wagner from the Department of Population Medicine, Harvard Medical School and Harvard Pilgrim Health Care Institute for insightful review and comments on this work.

## Author Contributions

**Conceptualization:** Yichen Zhang.

**Data curation:** Yichen Zhang, Dingyi Chen, Zhizhou Liang, Lin Bai.

**Formal analysis:** Yichen Zhang.

**Funding acquisition:** Xiaodong Guan.

**Investigation:** Yichen Zhang, Dingyi Chen, Siyuan Cheng, Zhizhou Liang, Lu Yang, Qian Li.

**Resources:** Luwen Shi, Xiaodong Guan.

**Supervision:** Luwen Shi, Xiaodong Guan.

**Validation:** Siyuan Cheng, Lu Yang, Qian Li, Wei Liu.

**Writing – original draft:** Yichen Zhang.

**Writing – review & editing:** Yichen Zhang, Huangqianyu Li, Xiaodong Guan.

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
