## [Editor Report · Decision Letter 0]

30 May 2023

Dear Dr Guan, 

Thank you for submitting your manuscript entitled "Use of Suboptimal Control Arms in Randomized Clinical Trials of Investigational Cancer Drugs in China, 2016-2021" for consideration by PLOS Medicine.

Your manuscript has now been evaluated by the PLOS Medicine editorial staff as well as by an academic editor with relevant expertise and I am writing to let you know that we would like to send your submission out for external peer review.

Please re-submit your manuscript within two working days, i.e. by Jun 01 2023 11:59PM.

Kind regards,

Philippa Dodd, MBBS MRCP PhD

Senior Editor

PLOS Medicine

---

## [Decision Letter · Decision Letter 1]

14 Sep 2023

Dear Dr. Guan,

Thank you very much for submitting your manuscript "Use of Suboptimal Control Arms in Randomized Clinical Trials of Investigational Cancer Drugs in China, 2016-2021" (PMEDICINE-D-23-01477R1) for consideration at PLOS Medicine. 

[LINK]

In light of these reviews, I am afraid that we will not be able to accept the manuscript for publication in the journal in its current form, but we would like to consider a revised version that addresses the reviewers' and editors' comments. Obviously we cannot make any decision about publication until we have seen the revised manuscript and your response, and we plan to seek re-review by one or more of the reviewers. 

We expect to receive your revised manuscript by Oct 05 2023 11:59PM. Please email us (plosmedicine@plos.org) if you have any questions or concerns.

We look forward to receiving your revised manuscript. 

Sincerely,

Philippa Dodd, MBBS MRCP PhD

PLOS Medicine

plosmedicine.org

GENERAL

Please respond to all editor and reviewer comments detailed below in full.

We recommend that you refer to our checklist of formatting requirements before resubmitting in order to expedite your manuscript through the technical checks. Download the formatting checklist (PDF).

TITLE

Please revise your title according to PLOS Medicine's style. Your title must be nondeclarative and not a question. It should begin with main concept if possible. "Effect of" should be used only if causality can be inferred, i.e., for an RCT. Please place the study design ("A randomized controlled trial," "A retrospective study," "A modelling study," etc.) in the subtitle (ie, after a colon).

COMPETING INTERESTS

All authors must declare their relevant competing interests per the PLOS policy, which can be seen here:

https://journals.plos.org/plosmedicine/s/competing-interests

For authors with ties to industry, please indicate whether any of the interests has a financial stake in the results of the current study.

ABSTRACT

Abstract methods and findings:

If not reporting p values for the purpose of transparent data reporting, please clearly state the reasons why not. Please report p values as <0.001 and where higher the exact as p value, p=0.002, for example.

Please include any important dependent variables that are adjusted for in the analyses.

Line 28 – please remove the word retrospective. Please check and amend throughout all subsections of the manuscript to refer to your study as, ‘observational’.

In the last sentence of the Abstract Methods and Findings section, please describe the main limitation(s) of the study's methodology.

Abstract Conclusions:

Suggest beginning, ‘In this study…’

The final statement could be more powerful, without over stating your results. Your data could have implications for clinical trial design and regulation of pharmaceutical companies. I think that’s what you mean by ‘collaborative efforts’ but it does not explicitly make the point and that is the real message.

AUTHOR SUMMARY

At this stage, we ask that you include a short, non-technical Author Summary of your research to make findings accessible to a wide audience that includes both scientists and non-scientists. The authors summary should consist of 2-3 succinct bullet points under each of the following headings:

• Why Was This Study Done? Authors should reflect on what was known about the topic before the research was published and why the research was needed.

• What Did the Researchers Do and Find? Authors should briefly describe the study design that was used and the study’s major findings. Do include the headline numbers from the study, such as the sample size and key findings. 

• What Do These Findings Mean? Authors should reflect on the new knowledge generated by the research and the implications for practice, research, policy, or public health. Authors should also consider how the interpretation of the study’s findings may be affected by the study limitations. In the final bullet point of ‘What Do These Findings Mean?’, please describe the main limitations of the study in non-technical language.

The author Summary should immediately follow the Abstract in your revised manuscript. This text is subject to editorial change and should be distinct from the scientific abstract. Please see our author guidelines for more information: https://journals.plos.org/plosmedicine/s/revising-your-manuscript#loc-author-summary

INTRODUCTION

Please indicate whether your study is novel and how you determined that.

If there has been a systematic review of the evidence related to your study (or you have conducted one), please refer to and reference that review and indicate whether it supports the need for your study.

Line 51 – ‘14% of RCTs…’ how many was this? It might be helpful to also include the approximate number of participants that these trials included (if known) such that the reader can get a clear picture of the extent of the issue.

Additional detail on how control arms are determined and by whom would be helpful. Does the problem exist at the level of the pharmaceutical company, or the regulators or both?

METHODS and RESULTS

We agree with the statistical reviewer, please see below, that investigation as to whether use of a sub-optimal control arm was associated with higher chances of concluding statistical significance would strengthen the manuscript.

Please ensure that percentages are quantified with numerators and denominators, not just percentages.

Where 95% CIs are reported, please also report values. Please report as p <0.001 and where higher as p=[exact p value]

Please separate upper and lower bounds of CIs with commas as opposed to hyphens as these can be confused with reporting of negative values.

DISCUSSION

Thank you for structuring your discussion according to PLOS Medicine’s style. Please remove all sub-headings from the discussion such that it reads as continuous prose.

TABLES

Table 1 – thank you for indicating that your analyses are adjusted. Please explicitly detail which factors are adjusted for in the caption. To help facilitate transparent data reporting, please also include the unadjusted analyses for comparison. Please report p as <0.001 and where higher the exact p value. Please define ‘CI’ for the reader (either the column header or the footnote). Please separate upper and lower bounds of 95% CIs with commas as opposed to hyphens as these can be confused with reporting of negative values.

Table 2 – please either define ‘No.’ or replace with ‘number’.

FIGURES

Figure 2 – please define ‘No.’ or use the word ‘number’.

DECLARATION STATEMENTS

Lines 305, 306 & 312 – please remove these statements and include only in the manuscript submission form when you re-submit the manuscript. In the event of publication these will be compiled as metadata.

REFERENCES

For in-text reference callouts please place citations in square brackets and preceding punctuation. For example, [1,3,6].’ Please check and amend throughout all subsections of the manuscript including the supporting information as relevant.

In the bibliography, please list up to but no more than 6 author names followed by et al. 

Please ensure any web references include an ‘Accessed [date]’

Journal name abbreviations should be those listed in the National Center for Biotechnology Information (NCBI) databases. 

Please see our website for other reference guidelines https://journals.plos.org/plosmedicine/s/submission-guidelines#loc-references

SUPPORTING INFORMATION

Please cite your Supporting Information as outlined here: https://journals.plos.org/plosmedicine/s/supporting-information

In the published article, supporting information files are accessed only through a hyperlink attached to the captions. For this reason, you must list captions at the end of your manuscript file. You may include a caption within the supporting information file itself, as long as that caption is also provided in the manuscript file. Do not submit a separate caption file.

eTable 2 and 3 – abbreviations for information detailed in the ‘Target’ column are not defined in the footnote (HER-2, for example). Please include definitions for all abbreviations.

Comments from the reviewers:

Reviewer #1: In this paper, authors assess the proportion of cancer RCTs in China that use suboptimal comparators by conducting a systematic search for RCTs using Chinese databases, and then assessing comparator against contemporaneous clinical practice guidelines (CPGs) in China. From a (saturation) sample of 454 trials, they find that about 14% of RCTs used suboptimal comparators. Trials recruiting only in China, as well as adjuvant and maintenance trials, were at elevated risk of using suboptimal comparators.

With some of the provisos laid out below I think this is an excellent manuscript addressing an important question. Why do trials if they use an inappropriate control? In my opinion the major cancer journals do not seem to want to answer this question (or reckon with the fact that this occurs). The manuscript is also very well researched and embeds the study in the a growing literature on comparator choice in cancer trials. Third, the study uses what are now considered well- established methods for assessing comparator quality. What makes this study unique and IMO potentially impactful is its focus on China- a major center for biomedical research- it is hard for those of us living in North America or Europe to get a good read on research practices in China. Also, the paper is written very clearly, and the results are transparently reported (with a good Supplementary Materials).

I offer the below comments and suggestions in the spirit of helping the authors refine the presentation and analysis.

- line 57- point about external validity is incorrect. The effect is still externally valid - comparative effects can be generalized to other settings. The problem is that the generalization is irrelevant since the comparison in the trial does not reflect current standards. Suggest rewording.

- Around line 57-65. I strongly feel that authors should appeal to basic ethical principles, including those expressed in Declaration of Helsinki or elsewhere, that say that patients should be provided best available therapy in comparator arms. Better yet would be an appeal to the principle of clinical equipoise, which is clearly violated when suboptimal comparators are used. In discussion, authors invoke ethical standards and Helsinki. But my feeling is that this should come earlier. It is simply unethical to randomize patients with life threatening diseases to suboptimal treatments, full stop. Any occurance of that means IRBs are not doing their job. Nor regulators. Nor physicians who agree to recruit their patients.

- Around line 99- why did authors only include adult trials? What proportion enrolled both adults and children?

- line 188-128- cross referencing NCCN and FDA approvals is a really nice methodological addition here.

- line 130: were extractions performed in duplicate and independently for eligibility and guideline matching? If so, what was the agreement rate? If not, can authors substantiate that there was enough agreement as to assure the reader of accuracy and reproducibility of extrations / matching?

- I am a little confused. I presume some trials in the sample were multicentric and international. So when counting numbers of patients, how did the authors deal with the fact that some patients may have originated from outside China (where standard of care might have varied or been deemed adequate)?

- Was this study prespecified in a protocol, with all hypotheses prespecified? If so, authors should name the database (e.g. Center for Open Science) and provide the registration record. If not, author should state that the study protocol and analyses were not pre-specified. Prespecification is increasingly the norm in meta-research of this type, though this reviewer acknowledges that many authors routinely bypass this fairly simple quality control measure.

- For optimal treatments, did authors assess dose or schedule of comparator arms? Sometimes studies use appropriate comparators but at a lowered dose, or on an inappropriate schedule. If authors did not do this, this should be stated as a limitation. Or better yet, author may want to assess this in their 86% of trials that used optimal compartors- or at least they might want to do this in a sample to see if it occurred.

- Line 181-3. This is tricky. With respect to category 4 of Box 1: Suppose two drugs are recommended, but one (A) is recommended at a higher level of evidence than the other (B). Would it not be the case that, if (B) is associated with less toxicity or easier administration (e.g. oral instead of IV), there might be sound reasons to recommend B rather than A (hence B IS a standard of care choice)? For combination- did authors check for simple substitutes of similar drugs, in which case while the exact combo is not the same, there are good reasons to think of the combo as equivalent to the one listed in the guidelines? [to some extent this is addressed around line 226 but I suspect this could be regarded as a controversial point]

- Totally up to authors, but they may want to cite (and link in to) my work on uninformative trials- in particular Zarin et al (JAMA 2019) and Hutchinson et al (eLife 2022).

- It's interesting that comparator choice seems more problematic with adjuvant and maintenance trials. Do authors have any thoughts on why this might be so? Might this relate to the point I made above about safety (a safer drug, taken over a more prolonged period but supported on weaker evidence, might be preferable to a more toxic drug supported by strong evidence).

- For readers unfamiliar with Chinese clinical practice guidelines, can authors say something about the quality of Chinese CPGs? For example, are there formal procedures used to evaluate and prioritize drugs for recommendations? Are they supported by systematic reviews? The strength of claims in this paper rests on the credibility of CPGs, so a comment (and evidence) on this would be very helpful.

- Beyond the scope of the article, but it would be great (and pretty easy, I think) to assess the proportion of trials that are positive on their primary outcome in the standard of care vs. suboptimal standard of care strata. A finding of a difference would underscore the impact of comparator bias. A null result would provide somewhat reassuring evidence that patients in comparator arms have not been harmed (though, of course, there is always the possibility that companies pursuing less promising drugs are more prone to picking weaker comparators to give their less promising drugs a boost).

- Also beyond the scope of the article: for multinational studies, authors consulted NCCN and FDA approvals. This affords authors to ask an additional question: to what extent were multinational trials enrolling in China prone to using comparators that were below standard of care in the U.S.? One would a priori predict that multinational studies enrolling in both U.S. and China would be "enriched" for use of suboptimal N American comparators, since such trials might under-accrue in the U.S. but more vigorously accrue in China. Totally optional (authors may want to see Awan et al Annals of Internal Medicine 2022 for a discussion of U.S. multinational cancer trials that also recruit in middle income countries).

- Also I wondered whether authors considered other subgroup analyses, including whether drug was combo vs. mono, whether company was large vs. small (biotech).

- In abstract and elsewhere (e.g. line 190), authors should report the cumulative number of patients at risk of entering a trial with suboptimal comparators in terms of proportions, not just absolute numbers.

- Fig 2, panel B. Please use other colours for bars. Very hard to distinguish these colours for many readers who are colour blind.

Very minor suggestions:

- Small grammar errors at line 188

- Line 195- report as p < some value (e.g. 001)

An additional thought- not necessary to address. The finding of 14% is pretty much in line with proportions reported using U.S. based or European trial samples. I think an additional insight this paper provides is that Chinese trials are no more misaligned with contemporary ethical standards than are U.S. or European studies- that is reassuring in a relative sense but discouraging in an absolute sense.

In short: timely study addressing an important question. Findings are important, work is original, methods are appropriate and transparent. I think there are things authors can do to improve the presentation, and they can report a bit more information on methodology so the reader can get a sense of quality checks in extraction and analysis. But all in all, this manuscript is promising. Apologies to authors (and editors) if there is anything crucial that I missed.

Jonathan Kimmelman

McGill University

Reviewer #2: Statistical review

This paper investigates the proportion of phase III cancer trials run in China that used a sub-optimal control arm (i.e. one that did not match treatment guidelines). The paper provides an interesting analysis. It uses (appropriately) straightforward analyses so there wasn't too much for me to comment on.

1. Page 5 - can the authors say more about the timing of when an arm was sub-optimal. If a trial started in (for example) 2017 using a control that was recommended in 2016 but not 2017 would that be classified as suboptimal? If standard of care changes during the course of the trial but the control arm is not updated, is the trial classified as using a suboptimal control arm?

2. Page 7 - can the authors provide the list of covariates considered in the model? Were these pre-specified?

3. Page 9 (and tables) 'p= 0.000' - I would report this as p<0.001 (but check PLOS guidance on p-values to make sure).

4. I'm not sure if the authors extracted information on whether the trial found a statistically significant result for the primary outcome. If so I would have found it interesting to see whether use of a sub-optimal control arm was associated with higher chances of concluding statistical significance.

James Wason

Reviewer #3: The authors present a retrospective, observational study to investigate the proportion of randomized trials of investigational cancer drugs that used a suboptimal control arm and the number of trial participants at risk of exposure to suboptimal treatments in China. These findings carry significant policy and clinical implications, as the use of a suboptimal control arm can hinder therapeutic innovation and compromise the benefits for participants. Overall, the methodology is sound and the manuscript is well-written. However, I have a few concerns that the authors should address before making definitive conclusions.

1.Some of the "suboptimal" control arms identified by the authors are questionable (eTable 3). For example, in the case of CTR20171559, it was actually conducted in the ≥3rd-line setting for treating advanced colorectal cancer, not the 2nd-line setting, making "FOLFOX/FOLFIRI/CapeOx±Cetuximab/Bevacizumab" not the guideline-recommended regimens in this context. Similarly, CTR20201983 was actually conducted in locally advanced esophageal cancer patients in the maintenance setting after CCRT, where "CCRT" is not the guideline-recommended regimen. It is also possible that the guidelines may not be updated timely to include the latest RCT results, leading to misclassified "suboptimal" control arms. For instance, CTR20212838 utilized SOX as the chemotherapy backbone in preoperative treatment for patients with locally advanced gastric cancer, which was supported by the positive results from the RESOLVE trial (doi: 10.1016/S1470-2045(21)00297-7) but not yet incorporated into the guideline at that time. As such, I would like to suggest the authors carefully re-assess the validity of the control arms of the included trials (if feasible, re-evaluation by oncologists in the field of specific cancer types).

2.The authors reported that the proportion of trials using a suboptimal control arm was substantially higher in China-only trials than in international trials (21.3% vs 8.3%). One potential reason for this disparity is that the authors reviewed NCCN guidelines only for international trials but not China-only trials, while some of the "suboptimal" control arms in China-only trials may be supported by NCCN guidelines. Therefore, I recommend re-evaluating the validity of the control arms in China-only trials by considering both CSCO and NCCN guidelines as references.

3.The authors utilized the level of recommendation from the CSCO guidelines as one of the criteria for defining a suboptimal control arm. However, it is worth noting that CSCO guidelines' level of recommendation takes into account various factors, including clinical efficacy data and whether the regimens are included in the medical insurance reimbursement list. It should be acknowledged that the latter criterion might not be reasonable for defining the validity of control arms if a trial did not violate the former criterion, particularly since the primary endpoint for registration RCTs is focused on efficacy rather than reimbursement considerations. 

4.The authors only included trials studying small-molecule entities and biologics. It would be helpful to clarify the reason for not including trials studying other drug types, such as chemotherapy.

[LINK]

---

## [Decision Letter · Decision Letter 2]

30 Oct 2023

Dear Dr. Guan,

Thank you very much for re-submitting your manuscript "Use of Suboptimal Control Arms in Randomized Clinical Trials of Investigational Cancer Drugs in China, 2016-2021: An Observational Study" (PMEDICINE-D-23-01477R2) for review by PLOS Medicine.

I have discussed the paper with my colleagues and the academic editor and it was also seen again by 3 reviewers. I am pleased to say that provided the remaining editorial and production issues are dealt with we are planning to accept the paper for publication in the journal.

[LINK]

If you have any questions in the meantime, please contact me on pdodd@plos.org or the journal staff on plosmedicine@plos.org.  

We look forward to receiving the revised manuscript by Nov 06 2023 11:59PM.   

Sincerely,

Philippa Dodd, MBBS MRCP PhD

PLOS Medicine

plosmedicine.org

COMMENTS FROM THE EDITORS

GENERAL

Thank you for your very detailed and considered responses to previous editor and reviewer comments. Please see below for further comments which we require that you address prior to publication.

The reviewer (please see below) notes that you report 84% and 100% of positive primary outcomes in RCTs which we agree is very high. We agree with the reviewer that this warrants further discussion as part of further revisions.

In line with the above the editorial team agree that discussion of how these data compare to the same/similar outside of China for cancer drugs and inside of China for other specialty meds (e.g Rheumatology) would be of value to better ascertain if this a country-specific issue or a specialty-specific issue – i.e. specific to cancer drugs. The Academic Editor also agrees, please see below.

COMMENTS FROM THE ACADEMIC EDITOR

Agree; the authors hint at the international comparison issue in the intro, but don't mention it again. Since this study is so laser-focused on China, would be useful to explain why they're focusing on China drug policy and how the Chinese FDA works/regulates in this area, at least as compared to FDA and EMA. Are these studies new studies done expressly for Chinese FDA approval, or are they the same pivotal studies submitted to other international regulators?

Line 26 – please remove the competing interest statement and include only in the masncuript submission form when you re-submit the manuscript. It will be compiled as metadata at the time of publication.

AUTHOR SUMMARY

Thank you for including an author summary.

Line 81 - What Did the Researchers Do and Find? Do you mean upward trend?

Line 90 – What Do These Findings Mean? Please make sentence beginning ‘Regulatory agencies…’ a separate bullet point.

REFERENCES

For in-text reference callouts citations should be placed in square brackets preceding punctuation. For example, line 100 should read, ‘for evaluating new drugs [1, 2].’ Please also note the addition of a space between the text and opening bracket. Please check an amend throughout all sub-section of manuscript.

SUPPORTING INFORMATION

Thank you for referencing and reporting yours according to PRISMA considering the search strategy applied to collection and extraction of your data. For completeness, please also include the PRISMA checklist as supporting as information. When completing the checklist, please use section and paragraph numbers, rather than page numbers.

Please add the following statement, or similar, to the Methods: "This study is reported as per the Preferred Reporting Items for Systematic Reviews and Meta-Analyses (PRISMA) guideline (S1 Checklist)."

Comments from Reviewers:

Reviewer #1: This paper continues to be a valuable contribution to the meta-research literature on randomized trials. It is well done, and address an important gap in the literature. The revisions address all of my concerns. There is one last point I want to raise- the authors may want to address it.

In revisions, the authors find that 84% and 100% of RCTs produced positive outcomes on their primary. That is just plain weird, and implies a lack of equipoise in the trials (on average, RCTs are positive 30-45% of the time; this reflects that answers of trials should be unknowable in advance of their conduct- see work of Benjamin Djulbegovic). To me, this high positive rate raises some concerns - perhaps orthogonal to the subject of the manuscript- about design or reporting practices used for this trial sample. I encourage authors- if they agree- to perhaps add a sentence or two in the discussion about this odd result.

Reviewer #2: Thank you to the authors for addressing my previous comments well. I have no further issues to raise.

Reviewer #3: I appreciate the authors' efforts on sufficiently addressing my concerns. I have no further comments.

[LINK]

---

## [Editor Report · Decision Letter 3]

6 Nov 2023

Dear Dr Guan, 

On behalf of my colleagues and the Academic Editor, Professor Aaron Kesselheim, I am pleased to inform you that we have agreed to publish your manuscript "Use of Suboptimal Control Arms in Randomized Clinical Trials of Investigational Cancer Drugs in China, 2016-2021: An Observational Study" (PMEDICINE-D-23-01477R3) in PLOS Medicine.

Prior to publication please also include reference to China’s ‘Guidance on Clinical Value-Oriented Oncology Drug Research and Development’ which was introduced in November 2021 in the introduction section of your manuscript. This will help to better ‘set the scene’ for the reader at the outset. 

PRESS

Thank you again for submitting to PLOS Medicine. It has been a pleasure handling your manuscript, we look forward to publishing your paper. 

Best wishes, 

Pippa

Philippa Dodd, MBBS MRCP PhD 

PLOS Medicine

pdodd@plos.org